# Detection of African Swine Fever at an Abattoir in South Korea, 2020

**DOI:** 10.3390/vetsci9040150

**Published:** 2022-03-22

**Authors:** Ki-Hyun Cho, Hyun-Joo Kim, Min-Kyung Jang, Ji-Hyoung Ryu, Daesung Yoo, Hae-Eun Kang, Jee-Yong Park

**Affiliations:** 1Foreign Animal Disease Division, Animal and Plant Quarantine Agency, 177 Hyeoksin 8-ro, Gimcheon 39660, Korea; vet10@korea.kr (K.-H.C.); mkjang0506@korea.kr (M.-K.J.); yjh562@naver.com (J.-H.R.); kanghe@korea.kr (H.-E.K.); 2National Institute of Health, Osong Health Technology Administration Complex, 187, Osongsaengmyeong 2-ro, Osong-eup, Heungduk-gu, Cheongju 28159, Korea; vetkhj@korea.kr; 3Department of Public Health, Graduate School, Korea University, 75 Goryeodae-ro, Seongbuk-gu, Seoul 02841, Korea; shanuar@korea.kr

**Keywords:** African swine fever, outbreak, pig farm, 2020, South Korea

## Abstract

In October 2020, a suspect case of African swine fever (ASF) was detected at an abattoir located in the north-central border region of South Korea. The farm of origin was traced and confirmed positive for ASF. This recurrence was following a period of absence of outbreaks in domestic pigs after the first incursion in 2019, during which a total of 14 domestic pig farms were confirmed between September and October 2019. In 2020, a total of two farms were confirmed, and the molecular characterization of key regions of the genome showed that the two isolates from 2020 were identical with the previous isolates from South Korea in 2019. The continued spread and circulation of ASF in the wild boar population represents an increased risk of spill-over outbreaks in domestic pigs, and, therefore, additional control measures should be implemented for farms in these regions, including a heightened level of surveillance. This was the case for the index farm, which was required to send pigs only to the designated abattoir at which the suspect case was quickly detected. The improvement of biosecurity in pig farms, particularly at the wild boar–domestic pig interface, will be key to the successful control of ASF in the region.

## 1. Introduction

African swine fever (ASF) is a devastating viral disease of swine species characterized by acute hemorrhagic fever and death. ASF is caused by the African swine fever virus (ASFV), which belongs to the family *Asfarviridae*, genus *Asfivirus*. ASF was first reported by Montgomery in Kenya (1921) and remains endemic in many sub-Saharan African countries. The first transmission of the disease outside of the African continent was by ASFV genotype I, which had spread to parts of Europe, South America, and the Caribbean, where it was later successfully eradicated, except for Sardinia, where it remains endemic to this day. In recent years, genotype II ASFV, which was first reported in Georgia in 2007, has shown a distinguished capacity for transboundary spread and has since been reported in numerous European and Asian countries. Notably, China reported its first ASF outbreak in a farm near Shenyang City in Liaoning Province in June 2018 [1], and it took only 9 months for the virus to disseminate across all the provinces of China.

South Korea, located in the southern part of the Korean peninsula in Northeast Asia, reported its first ASF outbreak on 16 September 2019, in a domestic pig farm in Paju City, Gyeonggi Province. The farm was located in the northwestern region of South Korea, approximately 7 km from the border with North Korea. A total of 14 ASF outbreaks were confirmed in domestic pigs, with the last being reported on 9 October 2019. All of the outbreak farms were located in the border region (Figure 1). The first ASF-infected wild boar was confirmed on 3 October 2019 within the fenced and heavily guarded 250 km long and 4 km wide demilitarized zone (DMZ) separating South Korea from North Korea [2]. The carcass was located at a distance of 8.4 km from the pig farm that would later be confirmed as the 14th ASF outbreak. Since then, wild boars infected with ASFV have been detected continuously in the northern border regions.

No further ASF outbreaks were reported in domestic pigs until a suspect case was detected in an abattoir in October 2020. Here, we report our findings, including the detection at the abattoir, the outbreak farm description, the molecular characterization of the virus, and the epidemiological situation in South Korea.

## 2. Case Presentation

### 2.1. Detection of Index Farm

On 8 October 2020, eight pigs that had arrived the day before at an abattoir in Cheorwon County, Gangwon Province, were found suspect for ASF. Initially, two sows were found dead during the morning ante mortem inspection at 08:30 am, while rest of the six sows from the same farm started showing symptoms of depression and recumbency. Another sow died soon after, at 10:50 am. Preliminary tests conducted by the Gangwon Province Veterinary Service Laboratory showed that the two dead sows were highly suspect for ASF.

The suspect pigs were traced to a farm in Hwacheon County, located next to Cheorwon County, where the abattoir is situated, which had been raising 721 pigs composed of 61 sows and 660 piglets. Clinical examinations were conducted by provincial veterinary service personnel, during which a dead sow was identified. Necropsy conducted at the farm showed that the pig had marked infarction and enlargement of the spleen, raising further suspicion of ASF. Immediately, blood samples from four pigs from the abattoir showing symptoms, and blood and tissue samples from the dead sow and a co-reared sow from the farm, were transported to the National ASF Reference Laboratory at the Animal and Plant Quarantine Agency (APQA) for confirmatory testing.

### 2.2. Laboratory Investigations and Confirmation of Additional Outbreak Farm

Submitted samples were received and tested in the Biosafety Level 3 (BSL-3) laboratory at APQA. Nucleic acids were extracted from homogenized organ samples and whole blood samples using Maxwell^®^ RSC Total Nucleic Acid kit and Maxwell^®^ RSC Whole Blood extraction kit (Promega, Madison, WI, USA), respectively, according to the manufacturer’s instructions. Real-time PCR was performed using methods described by King et al. [3]. Blood samples from the two pigs at the abattoir, and both the dead pig and the co-reared pig from the traced farm, were all confirmed ASFV-positive. All pigs in the infected farm were culled, during which additional blood samples from co-reared pigs and environmental swab samples were collected to assess the level of infection and degree of contamination at the farm [4]. Six of the twenty blood samples obtained from sows in the same pig house were found to be positive.

Two pig farms were located within the 10 km radius of the index farm and were preemptively culled. Blood and environmental samples were also collected from these farms and subjected to laboratory testing [4]. One of the two farms that was closer (2 km) to the index farm was also confirmed to be positive, with 2 of the 10 sows in the same farrowing house testing positive for ASF.

Hwacheon County, where both infected farms are situated, is located east of the northwestern border region where the 14 ASF outbreaks in domestic pig farms were confirmed in 2019, including Yeoncheon County, Paju City, Gimpo City, and Ganghwa County. Numerous cases of wild boars infected with ASF had been reported in Hwacheon County, including a case detected 230 m away from the index farm on 27 July 2020.

### 2.3. Genetic Characterization

Two ASFV strains were isolated in 2020 and designated as Korea/Pig/Hwacheon1/2020 and Korea/Pig/Hwacheon2/2020. Genetic analysis of the p72, intergenic region (IGR), p54, and EP402R gene sequences was conducted as described previously to identify and distinguish between closely related ASFV isolates [5]. Phylogenetic tree analysis showed that both ASFV isolates belonged to p72 genotype II (Figure 2a). In addition, these strains were of the IGR II variant, which has a 10 tandem repeat sequence (5′-GGAATATATA-3′) between I73R and I329L. Other strains that are also IGR II variants include the previous isolates from Korea in 2019, Ukr12/Zapo, Lt14/1490, and Pol14/Krus [5], and those from China (SY18 and CN201801) [6]. In addition, the nucleotide sequences of p54 from both strains were 100% homologous with those of Georgia 2007/1, SY18/China/2018, AnhuiXCGQ/China/2018, and VNUA/HY/Vietnam. The sequence of EP402R of both isolates are identical to those of the previous isolates from Korea in 2019, and Georgia 2007/1, AnhuiXCGQ/China/2018, China/Pig/HLJ/2018, and DB/LN/2018, which belong to serogroup 8 (Figure 2b).

## 3. Discussion

The two isolates were determined to be of the p72 genotype II, IGR II variant, and belonging to serogroup 8, as were the 14 isolates from 2019 [7]. ASFVs isolated from infected wild boars in South Korea are also of the same genotype and serotype, but diverse IGR variants (I, II, and III) have been reported [8]. The genetic analysis of the p72, CD2v, and IGR region between I73R and I329L provides valuable information, but not enough to trace the origin of infection or to confirm links between infection in wild boars and domestic pigs. However, the results support the view that genetically similar or identical ASFVs were responsible for the outbreaks in domestic pigs in 2019 and 2020, which could also be circulating in wild boars in South Korea. The whole genome sequencing of the Korean isolates from domestic pigs that is being conducted may provide further information.

The ASF outbreaks in October 2020 were promptly contained without further outbreaks. Effective ante mortem inspection conducted as part of the ASF surveillance scheme at the abattoir was a key factor in the early detection of ASF in 2020. Control measures, such as preemptive culling of the two farms within 10 km of the index farm and associated active surveillance, also contributed to the early detection of the additional infected farm and prevented further spread. The laboratory results of the two infected farms show that ASF infections at the time of detection were limited to only the pig housing with the infected pigs. This supports the view that ASF was detected in the early phase of infection. Epidemiological investigations to elucidate the source of infection for the 2020 outbreaks in domestic pigs are ongoing. However, the continued circulation and expansion of ASF in the wild boar populations, particularly those in the forested, mountainous areas of South Korea, pose a significant threat to the pig industry in South Korea. Forested mountains cover approximately 70% of the total land of South Korea, and they can serve as a good habitat for wild boars [9]. Notably, the Taebaek mountain range, which forms the main ridge of the Korean peninsula, stretches along the eastern coast, with two mountain chains extending to the west in the central region and the southwest of the central-southern region of South Korea (Figure 1). The national population density of wild boars was reported to be 4.1/km^2^ in October 2020, according to information released by MAFRA on 12 March 2021.

The swine industry accounts for 30% of the livestock sector in South Korea, being a highly intensive industry with a reported 11,784,312 heads being raised in 5725 farms in 2019. Domestic pig density is higher in the western part of the country and is focused in the central-western regions. The northern border regions where ASF outbreaks have been reported and the eastern mountain regions have a lower density [10]. However, further spread of ASF to the south or west, possibly through the mountain range, would place a far larger number of domestic pigs at risk, with greater socio-economic consequences.

Control measures for wild boars implemented by the Ministry of Environment include the elimination of potential sources of infection through carcass search and disposal, containment through fencing, and the reduction of the population by hunting and trapping [11]. Despite these efforts, ASF in wild boars continued to spread, and by 8 October 2020, a total of 758 cases were detected in nine counties/cities in the northern border region and its contiguous regions. The largest number of ASF cases in wild boars was reported in Hwacheon County (290 cases), where the two outbreak farms in 2020 are located, followed by Yeoncheon County (284 cases) and Paju City (98 cases). Regions with infected wild boars continue to expand eastward and southward along the eastern mountainous regions.

The geographical situation in South Korea, with its expansive mountainous areas, poses a major challenge when implementing control measures for wild boars by the relevant authorities. In addition, a recent assessment of the control measures implemented for wild boars listed biosafety problems and habitat- and management-related delays as hindrances to ASF control that will need to be addressed and better managed. For example, various instances were observed in the field when hunters and personnel involved in the search and disposal of carcasses did not observe proper biosecurity protocols [12].

During and after the outbreaks in domestic pigs in 2019, the Ministry of Agriculture, Food, and Rural Affairs (MAFRA) implemented control measures for domestic pigs, including the establishment of a buffer zone consisting of the infected region and its adjacent regions, with preemptive culling of all pig farms in this zone. In addition, all domestic pig farms in the four affected counties (Paju, Yeoncheon, Gimpo, and Ganghwa) in the northwestern border region were depopulated. For the remaining pig farms in the rest of the northern border regions, administrative orders were issued requiring laboratory tests prior to the movement of pigs, restriction of vehicle access, and regular monitoring of the sows. Sows from these regions were only allowed to be sent for slaughter at designated abattoirs, where strict clinical inspections and laboratory tests for suspected cases are performed as part of the ASF surveillance scheme. It can be regarded that the early detection of ASF in 2020 was, in part, possible due to the additional measures required for sows in the region.

## 4. Conclusions

Numerous pig farms in the northern regions are located in areas with active ASFV circulation in wild boars and are at a higher risk of ASF spillover from infected wild boars. Therefore, a coordinated effort by the relevant government bodies is needed to better manage this risk. One approach has been the provision of up-to-date information by the Ministry of Environment on the location of the positive wild boar cases, which is then used by MAFRA to designate the area within a 10 km radius of the positive case as an ASF control zone. Pig farms in such zones are placed under a higher alert state and are required to undergo additional measures when moving or sending pigs to the abattoir, including clinical examinations and laboratory testing. They are also required to send pigs only to designated abattoirs that take into account the geographical distance to minimize travel and reduce the possibility of ASF transmissions to non-infected regions, while requiring strict ante mortem inspections at arrival.

It is clear that the ASF epidemics in domestic pig and wild boar populations will continue to be influenced by the infection status of the other group, as spillover infection can occur in both ways and one population will continue to remain at a risk as long as the infection remains in the other. Therefore, control measures need to be effectively implemented for both populations through close and coordinated efforts by the relevant government bodies for the successful control of ASF in South Korea. This should include sharing experiences and the exchange of important information in real-time, as well as open communications between the ministries to harmonize and better coordinate government control activities.

## Figures and Tables

**Figure 1 vetsci-09-00150-f001:**
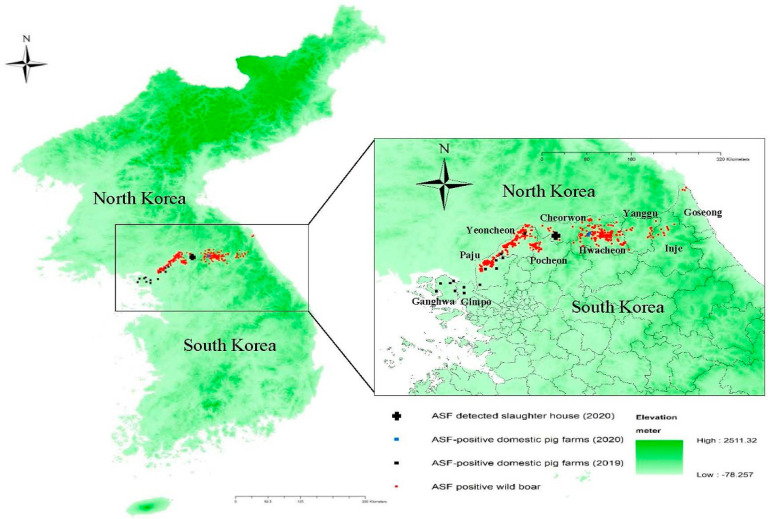
Map of African swine fever outbreaks in South Korea from 16 September 2019 to 10 October 2020, with the elevation of lands. The black rectangles and blue rectangles indicate ASF-positive domestic pig farms in 2019 and 2020, respectively. The red circles show the location of ASF-infected wild boars detected since 3 October 2019. The black cross indicates the location of the abattoir where ASF-infected pigs were detected in 2020. The degree of darkness of green on the map shows elevation of lands ranging from −78.28 to 2511.32 m from ground surface.

**Figure 2 vetsci-09-00150-f002:**
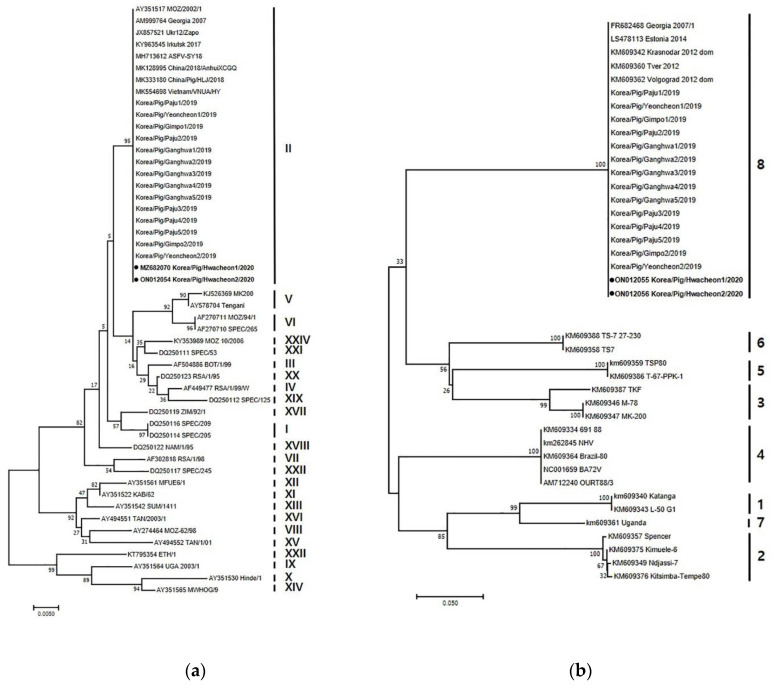
Phylogenetic analysis of ASFV isolated in domestic pig farms in South Korea during 2019-2020, based on partial sequence of the p72 gene (accession numbers MZ682070 and ON012054) and EP402R gene (accession numbers ON012055 and ON012056). (**a**) p72 genotype. (**b**) CD2v serogroup. The neighbor–joining method and Kimura 2-parameter model were used for constructing both phylogenetic trees in MEGA X software (https://www.megasoftware.net/). The two black dots indicate the ASFVs isolated in Korea in 2020.

## Data Availability

The data presented in this study are available on request from the corresponding author. The original contributions generated for the study are included in the article. Further inquiries can be directed to the corresponding author.

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
