# Peer review of "Detection of African Swine Fever at an Abattoir in South Korea, 2020"

_vetsci, 2022, doi:10.3390/vetsci9040150_

Round 1
Reviewer 1 Report
The authors present a clear description of the detection of ASFV in South Korea. The methods, results and conclusions are sound.
Suggested edits:
- Page 2, line 77 - change "homogenized" to "homogenized".
- Submit gene sequence data to Genbank and report accession numbers in the manuscript prior to publication.
- In the phylogenetic tree caption for Figure 2 please specify additional information on the phylogenetic methods used (program, phylogenetic method, and parameters).
Author Response
Thank you for the comments and suggestions. We have addressed the points as follows.
Point 1: Page 2, line 77 - change "homegenized" to "homogenized"
Response 1: The typo has been fixed to "homogenized".
Point 2: Submit gene sequence data to Genbank and report accession numbers in the manuscript prior to publication
Response 2: Accession number has been received for one of the isolate and has been added to the manuscript as "Two black dots indicate the ASFs isolated in Korea in 2020 (accession number MZ682070)".
We are however still waiting for the second number to arrive and will add accordingly.
Point 3. In the phylogenetic tree caption for Figure 2 please specify additional information on the phylogenetic methods used (program, phylogenetic method, and parameters) :
Response 3: The information has been added to the manuscript with the text "The neighbor-joining method and Kimura 2 parameter model were used for constructing both phylogenetic trees in MEGA X software (https://wwww.megasoftware.net/).
Reviewer 2 Report
The manuscript is well written and provides a detailed account of an outbreak of African swine fever involving two farms in South Korea. Although the scope and focus of the report is narrow, the concerning ASF situation globally means that contributions like this one add to the body of knowledge that we hope will be able to inform improved prevention of the disease. The time lapse between the first 14 outbreaks in domestic pigs in South Korea and these two outbreaks at a locality quite far from the first outbreaks is best explained by the large number of outbreaks in wild boars that occurred in a more or less continuous belt between the two outbreak areas. It is also of interest that the viruses from the two sets of domestic pig outbreaks are identical, while some variation at molecular level has been observed in wild boar isolates, a phenomenon also observed in eastern Europe. This report adds to our understanding of ASF and as such is definitely worth publishing, also because it is a high quality, well written investigation report.
There are some parallels with the situation in the Baltic States, in particular Latvia and Estonia, that you might like to check in the literature and refer to, but that is not compulsory!
Two minor comments:
Line 95: Delete one ‘cases’.
Figure 1: I hope that if the manuscript is published the symbols on the map will be a bit larger to enable one to distinguish them. I was not able to identify the cross marking the location of the abattoir at all, even at 150 x magnification on the computer
Author Response
Thank you for the comments and suggested improvements. The points have been addressed as follows.
Point 1: Line 95: Delete one ‘cases’.
Response 1: The sentence has been corrected as "Numerous cases of wild boars cases infected with..."
Point 2: Figure 1. I hope that if the manuscript is published the symbols on the map will be a bit larger to enable one to distinguish them.
Response 2: The symbols for the abattoir has been enlarged for better visualization.
Reviewer 3 Report
The authors have prepared a well written manuscript and the data supports their conclusions.
The only comment I have is that some discussion on different genotyping markers, e.g. those described in Poland and Vietnam is merited. There is also a full genome sequence from South Korea in GenBank. Does a comparison of the South Korean genome to others circulating in China justify the decision use to p54 and I73R-I329L IGS as subtyping tools?
Author Response
Thank you for the comments and the suggestions. We have addressed the points as follows.
Point 1. Some discussion on different genotyping markers, e.g. those described in Poland and Vietnam is merited.
Response 1. We have added the following description on the use of the specific genomes for differentiation between ASFV isolates. "Genetic analysis of the p72, intergenic region (IGR), p54 and EP402R gene sequences was conducted as described previously to identify and distinguish between closely related ASFV isolates [5]."
References
5. Gallaro, c. et. al